# Caveolae and Lipid Rafts in Endothelium: Valuable Organelles for Multiple Functions

**DOI:** 10.3390/biom10091218

**Published:** 2020-08-21

**Authors:** Antonio Filippini, Alessio D’Alessio

**Affiliations:** 1Department of Anatomy, Histology, Forensic Medicine and Orthopedics, Unit of Histology and Medical Embryology, Sapienza University of Rome, 00161 Roma, Italy; antonio.filippini@uniroma1.it; 2Dipartimento di Scienze della Vita e Sanità Pubblica, Sezione di Istologia ed Embriologia, Università Cattolica del Sacro Cuore, Fondazione Policlinico Universitario “Agostino Gemelli”, IRCCS, 00168 Roma, Italia

**Keywords:** angiogenesis, caveolae, caveolin, cavin, COVID-19, endocytosis, endothelial cells, lipid rafts, metabolism

## Abstract

Caveolae are flask-shaped invaginations of the plasma membrane found in numerous cell types and are particularly abundant in endothelial cells and adipocytes. The lipid composition of caveolae largely matches that of lipid rafts microdomains that are particularly enriched in cholesterol, sphingomyelin, glycosphingolipids, and saturated fatty acids. Unlike lipid rafts, whose existence remains quite elusive in living cells, caveolae can be clearly distinguished by electron microscope. Despite their similar composition and the sharing of some functions, lipid rafts appear more heterogeneous in terms of size and are more dynamic than caveolae. Following the discovery of caveolin-1, the first molecular marker as well as the unique scaffolding protein of caveolae, we have witnessed a remarkable increase in studies aimed at investigating the role of these organelles in cell functions and human disease. The goal of this review is to discuss the most recent studies related to the role of caveolae and caveolins in endothelial cells. We first recapitulate the major embryological processes leading to the formation of the vascular tree. We next discuss the contribution of caveolins and cavins to membrane biogenesis and cell response to extracellular stimuli. We also address how caveolae and caveolins control endothelial cell metabolism, a central mechanism involved in migration proliferation and angiogenesis. Finally, as regards the emergency caused by COVID-19, we propose to study the caveolar platform as a potential target to block virus entry into endothelial cells.

## 1. Introduction

The plasma (or cytoplasmic) membrane (PM) is likely the most crucial structure that safeguards and grants essential functions of eukaryotic cells. It separates the cytosolic compartments from the external environment, protects the cell from its surroundings, contributes to cell shape, controls cell-to-cell communications, and is central to transduce extracellular cues into specific biochemical processes. According to the fluid mosaic model proposed by S.J. Singer and Garth L. Nicolson in the early 1970s [1], the PM is a seemingly chaotic structure (i.e., a mosaic) built up with phospholipids, cholesterol, proteins, and carbohydrate residues that confer to the organelle its characteristic fluidity. It has been estimated that the protein:lipid ratio in cellular membranes is approximately 1:40 [2], indicating the importance of proteins in the organization and function of the phospholipid bilayer. In addition, the presence of a similar intramembranous system permits eukaryotic cells to compartmentalize specific metabolic activities into distinct cytosolic compartments. In certain cell types, the preferential segregation of glycosphingolipids and cholesterol within the exoplasmic leaflet of the PM [3,4] induces the formation of “liquid-ordered” less fluid domains termed lipid rafts (LRs) that are resistant to solubilization with nonionic detergents at low temperature [5,6]. This unique biophysical characteristic is crucial to separate the detergent-soluble membrane fractions (DSMs) from the detergent-resistant membrane fractions (DRMs) enriched in cholesterol, sphingolipids, and glycophosphatidylinositol (GPI)-anchored proteins [7]. Although there are numerous functions attributed to LRs in many cell types, the existence of these discrete plasma membrane microdomains in living cells remains elusive, mainly due to technical limitations that hinder their direct observation. The most used approach to study LRs takes advantage of their resistance to solubilization with detergents at 4 °C, and the high concentration of cholesterol observed is open to debate [8]. In addition, the formation of LRs has only been observed in artificial membranes, fomenting skepticism about the raft hypothesis in vivo [9]. LRs can further develop into 50–100 nm noncoated membrane indentations termed caveolae (from the Latin word “little caves”; singular, caveola), distinct from the electron-dense clathrin-coated pits [10]. By taking advantage of the emerging technique of electron microscopy, cell biologist George Palade was the first to report, in the early 1950s, the presence of a significant number of “plasmalemmal vesicles” in endothelial cells (ECs) facing the luminal surface of blood capillaries [11]. The term caveola was coined later by Enichi Yamada who observed analogous structures in the mouse gallbladder epithelium [12]. For about four decades, because of the lack of a definite molecular marker, caveolae were almost exclusively investigated at the morphological level by electron microscope, leaving wide open the debate regarding their biological significance. One of the well-known function of endothelial caveolae is transcytosis [13], which is the transport of macromolecules such as albumin, insulin, and LDL from the luminal side of the blood vessel to the subendothelial space—a mechanism that exhibits its greatest specialization in the blood–brain barrier of brain capillaries [14]. However, the discovery in 1992 of caveolin-1 (cav-1) [15] as the major scaffolding protein and a unique marker of caveolae, has raised increasing interest about the contribution of the caveolar network in cellular functions. Beyond its structural role in the biogenesis of caveolae, thanks to the presence of an oligomerization domain, cav-1 can influence protein membrane mobility and plasma membrane fluidity. More recently, a direct role of cav-1 in the organization of DRMs and the ability of the protein to induce liquid-ordered phase by direct acting on the phospholipid bilayer has been demonstrated [16]. Caveolae are considered subdomains of LRs [17] with a characteristic flask- (or omega-) shaped appearance. ECs, adipocytes, fibroblasts, and type I pneumocytes are particularly enriched in caveolae and express high levels of cav-1. The number of molecules associated with caveolae or interacting with cav-1 has rapidly increased and includes membrane receptors, G proteins-coupled receptors, non-receptors tyrosine kinase, enzymes, and structural proteins. The discovery of cav-1 and the generation of cav-1 knockout animals [18,19,20,21,22] have been crucial to deepen the role of the caveolar platform in signal transduction, cellular dysfunction, and vascular disease. Although cav-1 is known to be the major structural protein contributing to caveolae assembly, additional molecules, i.e., cavins, have been identified that take part in caveolar biogenesis [23,24,25,26]. The fundamental contribution of cavin members to caveolae biogenesis has been demonstrated in cavin1-knockout animals, which show an overall loss of caveolae in all tissues [27]. We will recapitulate here the most relevant and recent discoveries that have contributed to define the biological role of the caveolar platform in the physiopathology of the vascular endothelium.

## 2. Origin of ECs and Formation of the Vascular Three

At the time of implantation, the blastocyst is made of a thin outer covering, the trophoblast, and an inner cluster of cells termed the embryoblast. The first gives rise to a mitotically active layer termed the cytotrophoblast that further develops into the syncytiotrophoblast, an external epithelial layer without intercellular boundaries that actively invades the endometrial connective tissue and eventually surrounds the embryo. As the implantation moves forward, the embryoblast is subjected to morphological changes that lead to the transformation of the embryo into the so-called “bilaminar embryonic disc” composed of a thick epiblast (the floor of the amniotic cavity) and a thinner hypoblast (the roof of the exocoelomic cavity). During the third week, the establishment of the primitive streak on the surface of the epiblast signs the beginning of gastrulation, a fundamental process that generates the three germ layers, ectoderm, mesoderm, and endoderm. Specialized mesodermal precursors cells known as hemangioblasts, that, at first, appear in the primitive streak, migrate into the extra-embryonic yolk sac to form blood islands [28] (day 13–15 of gestation), composed of hematopoietic stem cells and angioblasts differentiating into red blood cells and ECs, respectively [29,30]. The differentiation and growth of primitive blood vessels from mesodermal-derived hemangioblasts, or vasculogenesis, is followed by a remodeling and expansion process during angiogenesis. Vascular endothelial growth factor (VEGF) represents one of the main factors that regulates the activities of ECs during both vasculogenesis and angiogenesis in the embryo as well as during neovascularization in adults. Additional data concerning the heterogeneity of blood vessel morphology have been recently reviewed [31]. Although the information about the presence of caveolin proteins in the embryo is limited and their expression levels change during embryogenesis, in mice embryos, all members of the caveolin family appear to be expressed for the first time between embryonic days E7 and E10 [10]. Notably, ECs of the developing lung have been shown to express the highest level of caveolin-1 mRNA in mice embryos, highlighting the potential role of this protein in lung vasculogenesis [32]. Studies aimed at depleting the expression of cav-1 in zebrafish embryos have highlighted the crucial role of cav-1α and -1β isoforms for the correct development of the nervous system, eye, somites, and cytoskeletal organization [33]. Although CAV-1^−/−^ animal models are viable and fertile, they exhibit reduced lifespan, organs hyperplasia, cardiomyopathies, vascular defects, and abnormal cellular functions [19]. These findings, the early expression of cav-1 in embryo development and the evolutionary conservation of caveolin genes, strongly support the crucial role of this protein in cell functions and vertebrate organogenesis.

### 2.1. The Caveolin Gene Family of Proteins

The discovery of cav-1, the first genuine marker of caveolae, has been decisive in the understanding of the biological significance of the caveolar network. Three members of the caveolin gene family, cav-1 (VIP21), cav-2, and cav-3 [34], have been identified with similar structure (Figure 1) and with variable level of expression between different organs and tissues in mammals [35,36,37]. All caveolin proteins are located on the cell surface and intracellularly in the Golgi apparatus, Golgi-derived vesicles, and endoplasmic reticulum (ER). A secreted form of cav-1 has been also identified both in normal and cancer cells [38,39]. Cav-1 and cav-2 are typically co-expressed and able to heterodimerize in a variety of cell types [40], particularly ECs, adipocytes, pneumocytes, and smooth muscle cells, where they appear to be mutually dependent for a proper cellular localization. Translation of the CAV1 gene located on chromosome 7q31.2 [41] produces a full-length protein of 178 amino acids of about 21kDa, known as cav-1α. A distinct mRNA [42,43] has been also identified that produces a shorter cav-1β isoform [44]. The full length cav-1 contains an oligomerization domain, spanning from residues 61–101, that serves to form high molecular mass homooligomers of about 350 kDa within the ER [45] that are trafficked to the Golgi apparatus and ultimately, delivered to the PM. A functional motif, termed caveolin scaffolding domain (CSD, residues 82–101) included into the oligomerization domains, promotes interactions with signaling proteins endowed with a caveolin binding motif [46,47,48,49] and is involved in tumor cell migration [50]. Cav-1 shows a peculiar topology with both the hydrophilic N- and C-termini projected into the cytoplasm and connected by a central hydrophobic domain (residues 102–134). The occurrence of mutations in the CAV1 gene are linked to severe diseases such as the congenital generalized lipodystrophy (CGL) [51,52], an autosomal recessive disorder characterized by the almost total loss of adipose tissue which generally concurs with severe hypertriglyceridemia, hepatic steatosis, splenomegaly, type 2 diabetes, and hypertension. Therefore, for all intents and purposes, the CAV1 locus adds to the list of other loci involved in the onset of lipodystrophies [53,54]. Recent studies have reported the contribution of cav-1 mutations to the onset of pulmonary arterial hypertension (PAH) [55,56,57,58], a rare and progressive degenerative disease that compromises the lungs vasculature. The altered production of NO in cav-1-deficient ECs [58] is crucial to vascular functions and this process is a critical candidate in the onset of PAH. The proper expression of cav-1 appears functional to maintain morphological features of ECs as well as to regulate TNF-induced signaling in ECs in vitro [59,60,61,62]. The role of cav-2, whose gene is located close to CAV1 on chromosome 7, has been less investigated. Although cav-1 and cav-2 form heterooligomeric complexes and show a similar intracellular distribution, the latter is unable (unlike cav-1) to form caveolae when overexpressed alone and is typically degraded in the absence of cav-1, indicating its principal role as a cav-1 stabilizer [19]. More recently, cav-2 has been linked to the progression of lung carcinoma [63] and the migration of pancreatic cancer cells [64], stimulating interest of this member of the caveolin family in pancreatic cancer. Cav-3, which is located on chromosome 3p25, exhibits high similarity to cav-1, but shows a restricted expression to all types of striated and smooth muscle cells [65]. Like cav-1, cav-3 protein serves as a scaffold to organize a complex molecular platform that regulates cell signaling in muscle cells. Several pathological conditions such as distal myopathy, hyperCKemia, rippling muscle disease, muscular dystrophy, and hypertrophic cardiomyopathy have been linked to cav-1 mutations in human and are now classified as caveolinopathies [66,67,68].

### 2.2. Caveolae Biogenesis and Membrane Dynamics

Caveolae are crucial flask-shaped and stripe-coated components of the PM that actively participate in several cell functions such as endocytosis, membrane trafficking, membrane dynamics, and signal transduction. However, the involvement of caveolae and LRs in these processes have been extensively revised in the literature and will be not further discussed in this review. It has been estimated that in adipocytes, up to 50% of the total amount of PM is represented by caveolae [69], while other cell types, such as erythrocytes and immune cells, appear to lack these structures [70]. Unlike the more well-known clathrin-coated vesicles, caveolae appear of smaller size, more stable, and less inclined to internalization [71,72,73]. Caveolae biogenesis starts with the synthesis of caveolin in ER, where it adopts the typical hairpin structure prior to moving to the Golgi complex, where it is palmitoylated and finally, is trafficked to the PM [74,75]. The delivery of caveolin from the Golgi apparatus to the cell surface is strictly dependent on cellular concentration of cholesterol [76]. Although caveolins are crucial during caveolae formation, other molecules contribute to their biogenesis. Members of the cavin family of proteins, including cavin-1, -2, -3, and -4 play an essential role in this process [77,78]. The four cavin proteins are encoded by different genes, namely, the polymerase I and transcript release factor (PDRF), serum deprivation response protein (SDPR), serum deprivation response factor-related gene product that binds to C-kinase (SRBC), and muscle-restricted coiled-coil protein (MURC) [77,79,80,81,82]. A cavin complex forms in the cytosolic compartment through the oligomerization of cavin subunits, reaches the cell membrane where it associates with caveolin, and contributes to sculpting caveolae [74,83]. While the role of cavin-1 and cavin-2 has been mainly related to caveolae stability and curvature [84], more recently, cavin-2 has been found to regulate crucial biological functions of different types of ECs, such as proliferation, migration, and in vitro angiogenesis by controlling the enzymatic activity of eNOS [85]. A recent study demonstrated that after stress-induced caveolae disassembly, the disengaged cavins establish new intercellular interactions related to a novel role for cavin proteins in metabolism and stress signaling [86].

### 2.3. The Role of Endothelial Caveolae in Mechanotransduction

By virtue of their position in the blood vessel wall, ECs in vivo are constantly exposed to hemodynamic forces resulting from blood flow, such as shear stress or those generated by blood pressure [87,88,89]. ECs are typically quiescent in physiological conditions but can rapidly turn into an active phenotype when exposed to aberrant blood flow, a condition that may lead to endothelial damage and dysfunctions [90,91] and whose outcome depends on the vessel type and location in the vascular tree. A major consequence of an aberrant blood flow is the generation of endothelial reactive oxygen species (ROS) through the activation of NADPH oxidase (NOX) enzymes, a process in which cav-1 plays a crucial role [92]. ECs withstand hemodynamic forces by activating their own mechanosensory complex [93,94] which is paramount to maintain vascular homeostasis and to prevent the onset of pathological conditions such as atherosclerosis [95,96,97]. Caveolae represent a disguised supply of cell membrane, whose cycling between disassembling and reassembly allows ECs to adapt to incoming mechanical cues [98,99]. In ECs, the mechanosome [100] includes structural constituents such as caveolae and molecules like PECAM-1, vascular endothelial growth factor receptor 2 (VEGFR2), vascular endothelial (VE)-cadherin, and probably others that function by converting physical forces into biochemical signals [101,102]. The contribution of caveolae in transducing shear stress into cells has been demonstrated [103,104], although the underlying molecular mechanisms have been only partially elucidated. Exposure of ECs to shear stress affects the distribution of cav-1, the amount of surface caveolae [105,106], and the activation of the ERK pathway [105,107,108], suggesting a key role of the caveolar platform to counteract changes in hemodynamic forces [99]. Similarly, other studies have suggested a role of caveolae and cav-1 in the sensing of hemodynamic forces that occur on the endothelial lining [104,109]. More recently, a meticulous flow analysis performed in bovine ECs has demonstrated the occurrence of a higher flow velocity above caveola than to its inside [110]. The authors demonstrated that neither VEGF-induced VEGFR2 phosphorylation nor VEGFR2 binding to cav-1 were affected by shear stress, indicating that caveolae may function as a platform where shear stress-sensitive receptors can safely bind their ligands while avoiding being exposed to shear stress [110]. However, these findings appear in conflict with a previous study reporting that surface level of VEGFR2 as well as VEGF-induced receptor phosphorylation were markedly increased in cells exposed to shear stress [111]. These contradictory results may perhaps be explained by the decreased number of caveolae that occurs in HUVECs [61] as well as other cell types in culture [112,113], hence, resulting in an underestimation of the effective role of caveolae in cultivated cells. More recently, it has been reported that the activity of YAP/TAZ, the major mediators of the Hippo pathway and among crucial components involved in mechanotransduction, depends on the presence of a functional caveolar platform [114,115] and contributes to EC sprouting [116,117,118]. While the increasing number of studies accumulated in the literature seems to confirm the role of the caveolae in mechanotransduction, further studies are necessary to elucidate the underlying molecular mechanisms. In particular, the use of in vitro models in caveolae depleted cells may be of help to investigate the precise sequence of events that characterize ECs response to flow and may contribute to elucidate the onset of cardiovascular conditions.

### 2.4. The Contribution of the Caveolar Platform to EC Metabolism

ECs that line the lumen of all blood vessels are known to be mainly quiescent [119,120,121] in physiological conditions, yet extremely able to adapt their proliferative and migratory status in response to angiogenic stimuli and growth factors such as VEGF, the most crucial player contributing to the formation of new blood vessels [122]. Whilst closely located to the highest available oxygen supply, i.e., the bloodstream, ECs prefer to rely on the less efficient aerobic glycolysis (the conversion of glucose into lactate instead of pyruvate) [123] rather than on oxidative phosphorylation (OXPHOS) for energy production [124]. It has been estimated that the number of mitochondria in the endothelium is considerably low with respect to other cell types, and in ECs, these organelles appear mainly involved in the regulation of cellular processes such as Ca^2+^ homeostasis, redox signaling, and cell death and only marginally contribute to EC energy needs [125,126]. By a mere energetic point of view, though the choice of ECs for the aerobic metabolism appears less advantageous with respect to the OXPHOS, it is functionally successful. Indeed, not only does this metabolic strategy reduce the production of ROS, due to the reduced amount of NADH that is re-oxidized in the mitochondria, but it also guarantees a reserve of oxygen to perivascular cells. Both proliferating and angiogenic ECs show a higher metabolic status compared to the resting ECs, as demonstrated by the elevated biosynthetic activity of ECs to become tip or stalk cells [127]. During the angiogenic switch ECs foster glycolysis for tip cell differentiation, whereas both glycolysis and mitochondrial respiration have been proposed to take place during proliferation of non-tip cells [128]. Metabolically, ECs appear like cancer cells that make the most of the glycolytic pathway for their energy requirements—a condition that was first described by Otto Warburg [129,130] and is known as the Warburg effect. Although, the relationship between tumor, ECs metabolism, and their angiogenic potential has been overlooked, these processes appear strictly related, as corroborated by the distinct metabolic conditions exhibited by quiescent and activated (angiogenic) ECs. Among the several enzymes that take part in the glycolytic process, 6-phosphofructo-2-kinase/fructose-2,6-bisphosphate 3 (PFKFB3), that is involved in the conversion of fructose-6-phosphate to fructose 1-6-bisphosphate, represents the rate-limiting factor for the whole glycolytic process [131,132]. Interestingly, PFKFB3 has been more recently linked the onset of cancer and its activity has been explored in many cancer cells types, whereas its contribution to the angiogenic potential of tumor-associated ECs (TECs) has been less investigated. To date, neither the metabolic changes that regulate the angiogenic switch nor the involvement of the caveolar network in the metabolic status of ECs have been fully elucidated. Although studies performed in cav-1 knockout mice revealed metabolic abnormalities [133], the underlying molecular mechanisms that link cav-1 and the caveolar network to EC metabolism are still poorly understood [134]. In adipocytes, caveolae directly participate in insulin signaling by sequestrating insulin receptor and GLUT4 and their perturbation leads to pathologic conditions related to lipid metabolism [135,136,137,138,139]. Notably, insulin resistance, defined as the lack of ability of cells to respond to insulin and to handle circulating glucose, is frequently associated with endothelial dysfunction and cardiovascular diseases. This is due, for example, to the deregulated interplay between insulin signaling and AKT-induced NO production in ECs, promoting atherothrombotic conditions [140,141]. In ECs, perturbation of caveolae biogenesis by the cholesterol binding agent filipin that specifically depletes caveolae but not clathrin-coated vesicles, demonstrated that a functional insulin receptor (IR) is required for insulin-induced cav-1 phosphorylation and insulin delivery to the muscle interstitium [142]. In addition, both filipin and the cholesterol extracting agent methy-β-lcyclodextrin inhibited insulin uptake in bovine aortic ECs [142]. More recently, it has been demonstrated that the Notch signaling plays a crucial role in the delivery of insulin from ECs to muscle cells by regulating gene expression associated with caveolae biogenesis [143]. These data not only confirm the active role of caveolae in the uptake of insulin, but also demonstrate the crucial role of endothelium in the delivery of both glucose and insulin across the vessel wall. Although the concentration of glucose in the blood is finely regulated by insulin, the circulating levels of fatty acids show greater variability in response to the nutritional food supply in human [144]. Therefore, endothelium has developed unique strategies to handle these unpredictable as well as potentially detrimental concentrations of fatty acids. It has been recently discovered that lipid droplets (LDs) [145] serve as fat storage from which ECs can either deliver fatty acids to the surrounding tissues or dispatch them back into the circulation [145]. However, although LDs can potentially represent a source of energy in the form of ATP, ECs, as mentioned before, mostly rely on glycolysis for energy production and the contribution of fats to this purpose is extremely modest in these cells. Interestingly ECs from cav-1^−/−^ exhibited reduced LD accumulation as compared to wild type animals, suggesting the contribution of cav-1 in the biogenesis of LDs in ECs [146]. In addition, cav-1 has been suggested to be proatherogenic thanks to its ability to deliver circulating LDL cholesterol to the subendothelial space. Caveolae-mediated transcytosis would function as an efficient carrier of cholesterol across the endothelial barrier, therefore, contributing to the initial stage of neointimal hyperplasia and atherosclerosis. To this regard, studies from cav-1 knockout mice demonstrated that cav-1 deficiency and the reduction in the number of caveolae play a protective role against atheroma formation, regardless of the presence of hypercholesterolemia [147,148,149]. In addition, reduced or total depletion of cav-1 strongly affects the extravasation of leukocytes through the endothelial layer, which is a crucial step in the initial development of the atherosclerotic plaque. This is mostly due to a reduced ability of leukocytes to adhere to ECs because of the reduced availability of CCL2 at the endothelial cell/monocyte interface and decreased VCAM expression seen in cav-1 knockout mice [150]. On the other hand, recent studies have suggested a role of cavins in the regulation of vascular remodeling, other than contributing to the shaping of caveolae. By employing a rat carotid artery balloon injury model, Zhou and collaborators demonstrated a marked reduction in cavin-1 expression in carotid arteries within two weeks after injury, suggesting the contribution of this protein in neointimal hyperplasia [151]. Interestingly, this effect of cavin-1 on increasing neointimal thickness was linked to caveolin-1 lysosomal degradation. Although further data are needed to deepen the precise molecular mechanisms, these data strongly confirm the interplay between caveolins and cavins during caveolae biogenesis, but as well, open new avenues of investigation related to vascular diseases.

### 2.5. Role of Caveolae/LRs in Virus Internalization by the Host Cell

Current general evidence suggests that the mechanism of virus cell entry could vary depending on viruses and host cell types, and can include clathrin-dependent endocytosis, caveolae/cholesterol endocytosis, and clathrin- and caveolae-independent mechanisms involving LRs [152,153]. The caveolae pathway occurs in enveloped viruses as well as non-enveloped viruses. A comprehensive review of the different viruses infecting various cells by the caveolae-mediated pathway has been recently reported [154]. It has been described that in the coronaviridae family, viruses such as HCoV-OC43 and HCoV-229E, both use caveolae for internalization, while SARS-CoV seems not to and prefers clathrin-mediated endocytosis, even though the role of cav-1 in SARS-CoV remains disputed. Overall, the pivotal event underlying SARS-CoV-2 infection is the endocytosis of viral particles, therefore, information on the mechanisms at the base of this intracellular membrane trafficking is crucial and may provide novel therapeutic opportunity to target specific molecules implicated in the key steps of the virus entry pathway. It has been reported that SARS-CoV can enter cells in the absence of clathrin-mediated endocytosis and interestingly, in cells infected with pseudovirus, virus entry was not inhibited by treatment with filipin or nystatin but, of note, was inhibited by methyl-β-cyclodextrin (MβCD), known to deplete cholesterol from the cell membrane, indicating that cholesterol- and sphingolipid-rich lipid raft microdomains act as a platform for virus entry [153]. ACE2 was identified as a functional receptor for SARS-CoV and relevant to the virus entry; Lu et al. [152] demonstrated that ACE2 is largely associated with LRs in Vero E6 cells [152] (Figure 2). Intriguingly, both the membrane proteases, Transmembrane Serine Protease 2 (TMPRSS2, essential for viral spread and pathogenesis) and Disintegrin and Metalloproteinase (ADAM17, also known as TACE) localize in LRs, and ADAM17 competes with TMPRSS2 to counteract virus cell entry by inducing the shedding of ACE2. Notably, both the functional activity and the surface distribution of TACE in ECs are dependent on the proper expression of cav-1 [155]. The fact that endocytic entry of SARS-CoV and, likely all coronavirus, involves lipid rafts, opens new avenues which could help to elucidate the intracellular itinerary of cell infection. Of note, the onset of cardiovascular complications secondary to respiratory distress that have been observed in coronavirus disease 2019 (COVID-19) would indicate the endothelium as a primary target of the virus. This is supported by signs of vascular dysfunctions seen in these patients such as increased blood pressure and thrombosis. ACE2 receptors are indeed expressed by ECs [156,157], where they have been found associated with LRs [152]. ECs, particularly pulmonary microvascular endothelial cells, are emerging as a primary target of SARS-CoV-2, although the link between vascular damage and COVID-19 is still unclear. When ECs are infected by viruses, they become more permeable and lose their capability to maintain vascular tone, a condition that is particularly evident when cells are exposed to inflammatory cues. The involvement of the caveolar network in EC inflammation and vascular tone is poorly explored. In addition, while postmortem analysis of patients revealed viral inclusions in ECs [158], the involvement of distinct PM domains of ECs related to vascular dysfunctions seen in COVID-19 is unclear. Therefore, the study of the role of the endothelial caveolar network in inflammation and coagulation and the identification of endothelial damage hallmarks such as abnormal levels of Angiopoetin-2, von Willebrand factor (vWF), thrombomodulin, and adhesion molecules, may open new avenues to develop novel therapeutic strategies in COVID-19 patients. Moreover, interfering with clathrin-independent endocytosis or disturbing the distribution of LR-resident molecules would be also of interest to study the mechanisms of SARS-CoV2 entry into the host cell.

## 3. Future Perspectives and Conclusions

Caveolae are present in the PM of many mammalian cell types and are particularly abundant in ECs. They have been involved in a wide range of processes including transcytosis, lipid homeostasis, cell metabolism, pathogen entry, angiogenesis, cancer progression, and cellular signaling. The discovery of integral proteins, such as caveolins and cavins, has been crucial to elucidate caveolae biogenesis and to investigate the role of these distinctive PM domains in vascular disease. The generation of caveolin knockout models as well as experimental evidence generated by RNAi on caveolins and cavin proteins markedly improved our understanding of the complex processes in which caveolae are involved. Although these studies have clarified that the absence of cav-1 does not affect fertility and is compatible with life, many adult diseases, including atherosclerosis, cellular transformation, and tumorigenesis, appear to be dependent on the aberrant expression of these scaffolding proteins. A great challenge that is still open is to distinguish between caveolar versus non-caveolar functions of cav-1 and cavins, which may be beneficial to develop novel therapeutic strategies in the field of vascular diseases. The emerging pandemic situation due to coronavirus syndrome made the investigation of pathogen entry into cells urgent. For this purpose, it would be interesting to discriminate the role played by LRs vs. caveolae and the specific role of caveolins during virus internalization into ECs. In addition, the discovery that the aberrant expression of cav-1 is linked to metabolic abnormalities and oxidative stress should be further investigated to deepen the molecular mechanisms leading to cardiovascular disease. Finally, beyond the undisputed role of cavins in caveolae shaping, the role of these proteins as well as cav-1 in signal transduction and the onset of atherosclerosis are less understood and deserve further investigation that may lead to the development of new clinical therapies for the treatment of vascular diseases. Therefore, we can expect that the continuous investigation of caveolae-related processes and the understanding of the role played by the plethora of caveolin interacting molecules, are likely to provide novel details concerning the significance of this organelle in health and disease.

## Figures and Tables

**Figure 1 biomolecules-10-01218-f001:**
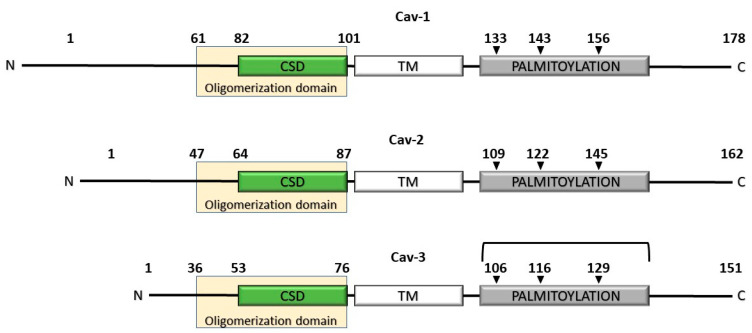
The human caveolin gene family. The indicated palmitoylation sites of cav-3 have been assumed on sequence alignment with cav-1, but they have not been experimentally determined yet.

**Figure 2 biomolecules-10-01218-f002:**
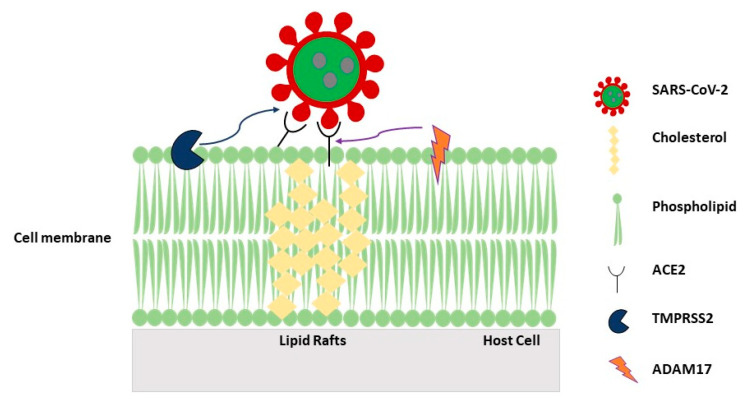
Scheme of coronavirus SARS-CoV-2 entry into the host cell. Virus uses angiotensin-converting enzyme 2 (ACE2) to bind the host cell and the cellular protease TMPRSS2 for viral entry. Cleavage of ACE2 ectodomain by ADAM17 residing in lipid rafts may contribute to regulate availability of receptor-mediated endocytosis of virus.

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
