# Peer review of "Caveolae and Lipid Rafts in Endothelium: Valuable Organelles for Multiple Functions"

_biomolecules, 2020, doi:10.3390/biom10091218_

Round 1

Reviewer 1 Report

Current manuscript  by Antonio Filippini and Alessio D’Alessio comprise well-made comprehensive review into caveolar function in endothelium. In general, it is well-written and covers most studies in this field, focusing mostly on virus internalization and mechanotransduction of this tissue-specific subtype of caveolae. As caveolae were also shown to be involved on other functions such as signal transduction and transcytosis/endocytosis, it will be useful to mention that authors chooses to cover only part of these proposed functions in this review.

Author Response

Author’s response (indicated in italic) to Reviewer #1:

Current manuscript by Antonio Filippini and Alessio D’Alessio comprise well-made comprehensive review into caveolar function in endothelium. In general, it is well-written and covers most studies in this field, focusing mostly on virus internalization and mechanotransduction of this tissue-specific subtype of caveolae. As caveolae were also shown to be involved on other functions such as signal transduction and transcytosis/endocytosis, it will be useful to mention that authors chooses to cover only part of these proposed functions in this review.

We thank the reviewer for her/his valuable comments and appreciation of our work. It was not our aim to recapitulate the role of caveolae and lipid rafts in endocytosis and signal transduction, as these topics have been widely reviewed by many authors. We have clearly stated this intention both in the abstract section and at line 192 in the paragraph titled “Caveolae biogenesis and membrane dynamics “, at page 7 of the revised manuscript.

Reviewer 2 Report

  1. Define the abbreviation LR on line 63.
  2. On lines 59-63, DRMs are discussed.  Additional sentences are needed to discuss the controversy surrounding this technique and its lack of clear benchmarks.
  3. Line 152, change "therefore, to" to "therefore, for".
  4. Sentence starting on line 157 needs to be rewritten, it is not clear.
  5. Figure 1, the box for palmitoylation residues should be removed and the exact palmitoylated residues shown. See PMID 7896831 for Cav1 and PMID 25667086 for Cav2.  It is important to note that while Cav3 is palmitoylated the exact sites are not known (PMID 10464299).
  6. The "Future perspectives and conclusions" section should be rewritten it is too vague.  The authors should clearly identify several emerging areas related to caveolin biology and explain why they are important to investigate.

Author Response

Author’s response (indicated in italic) to Reviewer #2:

Define the abbreviation LR on line 63.

The abbreviation LRs has been defined at line 61 of the revised manuscript as requested.

On lines 59-63, DRMs are discussed. Additional sentences are needed to discuss the controversy surrounding this technique and its lack of clear benchmarks.

We thank the reviewer for highlighting the methodological issue regarding the detection of lipid rafts domains. We have included few more lines (65-72) in the revised manuscript, including two new references N.8 and 9,to discuss this topic. We hope it fully answers the reviewer's observation.

Line 152, change "therefore, to" to "therefore, for".

We have corrected the text as suggested by the reviewer. See line 166 of the revised manuscript.

Sentence starting on line 157 needs to be rewritten, it is not clear.

We agree with the reviewer that this sentence is inappropriate. We decided to delete the entire sentence from the text and move the reference (56) above. See line 172 of the revised manuscript.

Figure 1, the box for palmitoylation residues should be removed and the exact palmitoylated residues shown. See PMID 7896831 for Cav1 and PMID 25667086 for Cav2.  It is important to note that while Cav3 is palmitoylated the exact sites are not known (PMID 10464299).

We thank the reviewer for giving us the opportunity to delve into this topic. We would like to take the reviewer's attention to the fact that there is some confusion in the literature about the exact palmitoylation sites of caveolins. However, concerning cav-3 palmitoylation sites, we referred to a relatively recent publication (see new reference 36) where the palmitoylation sites of this protein are clearly indicated. We have accordingly revised Figure 1 as suggested by the reviewer by removing the palmitoylation boxes, leaving only arrowheads indicating the palmitoylated residues. In addition, figure 1 has been graphically modified to improve its readability. Finally, three more relevant references (35, 36 and 37) referring to this topic have been included in the revised version of the manuscript. We hope these changes can fulfill the reviewer's comment.

The "Future perspectives and conclusions" section should be rewritten it is too vague. The authors should clearly identify several emerging areas related to caveolin biology and explain why they are important to investigate.

We thank the reviewer for her/his valuable suggestion. We have accordingly revised and expanded the paragraph "Future perspectives and conclusions" to meet the reviewer’s suggestion.

Reviewer 3 Report

The work is interesting and well structured.

Minor points to review:
1) in the introduction it would be interesting to add biophysical studies showing the role of caveolin in the organization of the ordered liquid phase.
2) to make the work more homogeneous it would be appropriate to move "Role of caveolae / LRs in virus internalitazion by the host cell" as last before the conclusions.
3) in the paragraph "the contribution of caveolar platform to EC metabolism" the contribution of caveolae in insulin signaling is described in detail, however its role in the progression of atherosclerosis described in the last part should be better explained also analyzing the role of the cavin proteins

Author Response

Author’s response (indicated in italic) to Reviewer #3:

In the introduction it would be interesting to add biophysical studies showing the role of caveolin in the organization of the ordered liquid phase.

We added a new sentence regarding the role of caveolin in the organization of the ordered liquid phase including a new reference n.16. See lines 88-92 of the revised manuscript.

To make the work more homogeneous it would be appropriate to move "Role of caveolae / LRs in virus internalization by the host cell" as last before the conclusions.

We have reorganized the manuscript as suggested by the reviewer. Therefore, in the revised version of the manuscript, the paragraph "Role of caveolae / LRs in virus internalization by the host cell" is the last one before “Future perspectives and conclusions”.

In the paragraph "the contribution of caveolar platform to EC metabolism" the contribution of caveolae in insulin signaling is described in detail, however its role in the progression of atherosclerosis described in the last part should be better explained also analyzing the role of the cavin proteins

We thank the Reviewer’s suggestion and we have modified this section of the manuscript accordingly. The new part in the paragraph is indicated with blue text.

Round 2

Reviewer 2 Report

1. The added sentences starting on line 43 need better grammar. Below are my edits.

"Although there are numerous functions attributed to LRs in many cell types, the existence of these discrete plasma membrane microdomains in living
cells remains elusive mainly due to technical and limitations that hinder their direct observation. The most used approach to study LRs takes advantage of their resistance to solubilization with detergents at 4°C, and the high concentration of cholesterol observed is that are often open to debate misinterpretation [8]. In addition, the formation of LRs have has only been only observed in artificial membranes, fomenting skepticism about the rafts hypothesis in vivo [9]."

2.  Yes, the authors are correct that reference 36 shows the palmitoylation sites for caveolin-3, but that paper is incorrect.  The authors in that paper cite a book which states that based on sequence alignment with caveolin-1, it is assumed that caveolin-3 is palmitoylated at the same positions but the definitive palmitoylation sites for caveolin-1 have not been determined.

Author Response

1. We apologize with the reviewer for the inaccuracy of the sentence at line 43. It has been accordingly corrected as suggested.

We accurately revised the references reported in the previous version of the manuscript. We agree with the reviewer that there are some inaccuracies in the literature related to the palmitoylation sites of cav-3.

We have clearly stated in the caption of figure 1 that the palmitoylation sites of cav-3 have only been assumed on the sequence alignment with cav-1. However, they have not been fully determined.

We thank the reviewer for this comment and for giving us the opportunity to add in our manuscript the most accurate details regarding this topic.

Figure 1. The human caveolin gene family. (?) The indicated palmitoylation sites of cav-3 have been assumed on sequence alignment with cav-1 but they have not been experimentally determined yet.

This manuscript is a resubmission of an earlier submission. The following is a list of the peer review reports and author responses from that submission.